# Controversial Aspects of Diagnostics and Therapy of Arthritis of the Temporomandibular Joint in Rheumatoid and Juvenile Idiopathic Arthritis—An Analysis of Evidence- and Consensus-Based Recommendations Based on an Interdisciplinary Guideline Project

**DOI:** 10.3390/jcm11071761

**Published:** 2022-03-22

**Authors:** Christopher Schmidt, Rudolf Reich, Bernd Koos, Taila Ertel, Marcus Oliver Ahlers, Martin Arbogast, Ima Feurer, Mario Habermann-Krebs, Tim Hilgenfeld, Christian Hirsch, Boris Hügle, Thekla von Kalle, Johannes Kleinheinz, Andreas Kolk, Peter Ottl, Christoph Pautke, Merle Riechmann, Andreas Schön, Linda Skroch, Marcus Teschke, Wolfgang Wuest, Andreas Neff

**Affiliations:** 1Department of Oral and Craniomaxillofacial Surgery, UKGM GmbH, University Hospital Marburg, Faculty of Medicine, Philipps University, 35043 Marburg, Germany; schmidt-cb@web.de (C.S.); taila.ertel@t-online.de (T.E.); merlee.riechmann@t-online.de (M.R.); l.skroch@gmx.de (L.S.); 2Practice for Oral and Maxillofacial Plastic Surgery MVZ R(h)einZahn, 53111 Bonn, Germany; rh-reich@t-online.de; 3Department of Orthodontics, University Hospital Tübingen, Eberhard Karls University Tübingen, 72074 Tübingen, Germany; bernd.koos@med.uni-tuebingen.de; 4Medical Clinic, Hospital of Winsen, 21423 Winsen (Luhe), Germany; 5Department of Prosthetic Dentistry, Center for Dental and Oral Medicine, University Hospital Hamburg-Eppendorf, University of Hamburg, 20146 Hamburg, Germany; ahlers@uke.de; 6CMD-Center Hamburg-Eppendorf, 20251 Hamburg, Germany; 7Department of Rheumatic Orthopedics and Hand Surgery, Hospital Oberammergau, 82487 Oberammergau, Germany; martin.arbogast@wz-kliniken.de; 8Physiotherapeutic Practice & Orthopedic Manual Therapy, 78315 Radolfzell-Böhringen, Germany; ima.feurer@t-online.de; 9Deutsche Rheuma-Liga Bundesverband e.V. (German Patients’ Association for Rheumatic Disorders), 53111 Bonn, Germany; habermann-krebs@gmx.de; 10Department of Neuroradiology, University Hospital Heidelberg, Ruprecht-Karls-University Heidelberg, 69120 Heidelberg, Germany; tim.hilgenfeld@med.uni-heidelberg.de; 11Clinic of Pediatric Dentistry, University Hospital Leipzig, University of Leipzig, 04103 Leipzig, Germany; christian.hirsch@medizin.uni-leipzig.de; 12German Centre for Pediatric Rheumatology, Childrens’ Hospital Garmisch-Partenkirchen, 82467 Garmisch-Partenkirchen, Germany; huegle.boris@rheuma-kinderklinik.de; 13Department of Pediatric Radiology, Olgahospital, Klinikum Stuttgart, 70174 Stuttgart, Germany; t.vonkalle@klinikum-stuttgart.de; 14Department of Cranio-Maxillofacial Surgery, University Hospital Münster, 48169 Münster, Germany; johannes.kleinheinz@ukmuenster.de; 15Department of Oral and Craniomaxillofacial Surgery, University Hospital Innsbruck, Leopold-Franzens-University Innsbruck, 6020 Innsbruck, Austria; andreas.kolk@i-med.ac.at; 16Department of Prosthodontic Dentistry, University Hospital Rostock, University of Rostock, 18057 Rostock, Germany; peter.ottl@med.uni-rostock.de; 17Medical Practice & Clinic for Oral and Maxillofacial Surgery, 80333 Munich, Germany; christoph.pautke@gmx.net; 18Medical Practice & Clinic for Oral and Maxillofacial Surgery, 53842 Troisdorf, Germany; andreas.schoen@ukbonn.de; 19Medical Practice for Oral and Craniomaxillofacial Surgery, 61352 Bad Homburg, Germany; marcus.teschke@me.com; 20Children’s Hospital Wilhelmstift, 22149 Hamburg, Germany; 21Department of Radiology, Hospital Martha Maria Nürnberg, 90491 Nuremberg, Germany; wolfgang.wuest@martha-maria.de

**Keywords:** chronic rheumatic arthritis of the temporomandibular joint, rheumatoid arthritis, juvenile idiopathic arthritis, guideline, Delphi method, consensus

## Abstract

Introduction: Due to potentially severe sequelae (impaired growth, condylar resorption, and ankylosis) early diagnosis of chronic rheumatic arthritis of the temporomandibular joint (TMJ) and timely onset of therapy are essential. Aim: Owing to very limited evidence the aim of the study was to identify and discuss controversial topics in the guideline development to promote further focused research. Methods: Through a systematic literature search, 394 out of 3771 publications were included in a German interdisciplinary guideline draft. Two workgroups (1: oral and maxillofacial surgery, 2: interdisciplinary) voted on 77 recommendations/statements, in 2 independent anonymized and blinded consensus phases (Delphi process). Results: The voting results were relatively homogenous, except for a greater proportion of abstentions amongst the interdisciplinary group (*p* < 0.001). Eighty-four percent of recommendations/statements were approved in the first round, 89% with strong consensus. Fourteen recommendations/statements (18.2%) required a prolonged consensus phase and further discussion. Discussion: Contrast-enhanced MRI was confirmed as the method of choice for the diagnosis of TMJ arthritis. Intraarticular corticosteroid injection is to be limited to therapy-refractory cases and single injection only. In adults, alloplastic joint replacement is preferable to autologous replacement. In children/adolescents, autologous reconstruction may be performed lacking viable alternatives. Alloplastic options are currently still considered experimental.

## 1. Introduction

A variety of rheumatic underlying diseases may lead to a TMJ involvement as they progress. A differentiation is made between juvenile idiopathic arthritis (JIA), by definition with onset prior to 16 years of age, and rheumatic arthritis in adulthood, with rheumatoid arthritis (RA) most prevalent amongst them [1,2]. The disease termed chronic rheumatic arthritis of the temporomandibular joint may manifest in synovitis, bone alterations and deformation, inhibited growth in children, up to complete condylar resorption, and ankylosis of the TMJ [3]. However, standardized terminology has been lacking for inflammations affecting the TMJ in the context of a chronic rheumatic underlying disease. In 2019, an international consensus achieved by the TMJaw working group on definitions was published for the first time. For this purpose, TMJ arthritis is defined to denote exclusively an active inflammation of the TMJ, while TMJ involvement is defined to include any anomalies presumably resulting from TMJ arthritis [4].

The inconsistent correlation between the results of imaging and clinical symptoms is a specific challenge in diagnosing TMJ arthritis. While indications may point to substantial damage to the joint, patients may otherwise have few complaints or may even be asymptomatic [5,6]. Based on a reported prevalence between 45% and 92% for RA and 40% to 93% for JIA [7], Arabshahi and colleagues concluded that the frequency of the involvement of the TMJ has long been underestimated [8]. Some authors even rate the TMJ as one of the most frequently affected joints in JIA [9,10].

Considering the potential grave complications listed earlier, early diagnosis and onset of therapy of arthritis of the temporomandibular joint are of vital importance. Despite considerable international efforts, e.g., in the framework of TMJaw and OMERACT working groups [4,11,12,13,14,15], sufficient evidence and standardized approaches are still lacking in many areas of clinical management, from which derives the significance of the interdisciplinary, evidence- and consensus-based guideline, which was published online in Germany by the Association of the Scientific Medical Societies in Germany (Arbeitsgemeinschaft der Wissenschaftlichen Medizinischen Fachgesellschaften—AWMF) as an evidence- and consensus-based S3 guideline in July 2021. The aim of this study is to identify and analyse the issues subject to controversial discussion during the development of the above S3 guideline, based on a systematic literature search and an interdisciplinary expert consensus of mandated representatives of the contributing scientific societies.

## 2. Materials and Methods

### 2.1. Preparation of Project

After the guideline project was approved by the guideline officer and the board of directors of the German Association for Oral and Maxillofacial Surgery (Deutsche Gesellschaft für Mund-, Kiefer- und Gesichtschirurgie—DGMKG), the project was registered with the Association of the Scientific Medical Societies in Germany (Arbeitsgemeinschaft der Wissenschaftlichen Medizinischen Fachgesellschaften—AWMF). The registration of the project was published online on the AWMF website under registration number 007-061. At first, the members of the temporomandibular joint surgery (TMJS) guideline group of the DGMKG were contacted. A TMJS panel was formed from members who agreed to participate (see Appendix A), with the aim of compiling a preliminary draft based on a systematic literature search (see Section 2.2). At the second stage, relevant medical societies involved in the subject and patient organisations were specifically selected and invited to participate in the guideline project. An interdisciplinary working group was formed from the mandated reresentatives of the participating medical societies (see Appendix A), and specific medical and technical questions were further explored in subgroups. Any potential conflicts of interest were investigated by the steering committee (A.N., Ch. S. und T.E.) and could be successfully ruled out.

As a guideline for the systematic literature search and a structured data acquisition, the PRISMA 2020 Checklist (acronym: Preferred Reporting Items for Systematic Reviews and Meta-Analyses) was used [16]. Applied items of the checklist are mentioned in the associated methodical segments in the text below.

### 2.2. Systematic Literature Search

Due to the absence of thematical relevant guidelines and the limited level of evidence the guideline working group decided against the initial formulation of a specific research question using the PICOTS framework by the Agency for Healthcare Research and Quality [17] (acronym: P: patient population, I: intervention, C: comparator, O: outcome, T: timing, S: setting). Rather, a broader search strategy was applied to define patients with a TMJ affection in the context of rheumatic disease as the target population. Specifications on interventions, control groups, outcome variables, follow-up periods and healthcare settings were initially avoided due to before mentioned reasons (PRISMA item 4). Therefore, the literature search was conducted using the syntax: “temporomandibular joint [AND] rheumatoid arthritis [OR] juvenile idiopathic arthritis [OR] psoriatic arthritis [OR] ancylosing spondylitis”. The use of German or English language was defined as an inclusion criterion. Studies researching subjects other than human beings (e.g., animal studies, cadaver studies, laboratory studies and finite element studies) were excluded. The guideline working group decided against limiting the search to specific years of publication or study types due to the before mentioned limited level of evidence. Specific research questions arising from the initial search were researched using the PICOTS framework (PRISMA items 5 and 7).

Initially, a search for national and international guidelines was conducted within the databases PubMed and Cochrane, the AWMF database, www.guideline.gov (accessed on 15 February 2022), www.nice.org.uk (accessed on 15 February 2022) and the websites of the European (EACMFS: European Association of Cranio-Maxillo-Facial Surgery, https://www.eacmfs.org/, accessed on 15 February 2022) and American (AAOMS: American Association of Oral and Maxillofacial Surgeons, https://www.aaoms.org/, accessed on 15 February 2022) specialist medical societies for oral and maxillofacial surgery (OMFS). This was followed by a search for individual publications in PubMed, PubMed Central (PMC), Embase, Cochrane Library und Livivo databases (PRISMA item 6). Titles and abstracts of potentially eligible records were reviewed and assessed by two medical experts independently (C.S. and T.E.). Duplicates and studies not fulfilling the inclusion criteria or fulfilling the exclusion criteria were removed. A further selection was made by removing publications with non-suitable target populations based on a full text analysis. Additional sources were included in the guideline after an additional manual search and following a literature update (Figure 1) (PRISMA items 7, 8 and 9).

### 2.3. Assessment of Evidence

The Level of Evidence (LoE) was assessed by two medical experts (T.E. and C.S.) independently from each other, based on the Oxford Criteria [18] (see Table 1). In case of doubt, a third expert was consulted (A.N.) (PRISMA item 9).

The methodological quality was assessed according to the SIGN-Checklists (http://www.sign.ac.uk/checklists-and-notes.html, accessed on 15 February 2022) by two experts independently (C.S., T.E.) and classified into four categories respectively (see Table 2). In case of fundamental disagreement between the experts, a third expert was consulted (A.N.) (PRISMA item 11).

With SIGN checklists available as a tool for assessment of methodological quality for Levels of Evidence Ia to IIIb only, and due to the widely heterogeneous quality of studies of Level of Evidence IV and V, an additional assessment of clinical relevance was introduced and applied to all study types and evidence levels. Criteria applied were, depending on the study subject and methodology: patient sample size, relevance of research question and target numbers, disclosure of patient characteristics, inclusion and exclusion criteria, duration of follow-up, “lost-to-follow-up” rates and suitability of comparison intervention and control group. Assessments were made in a similar manner to SIGN checklists by two independent experts (C.S., T.E.) as outlined in Table 3. In case of doubt, a third expert was consulted (A.N.) (PRISMA item 11).

### 2.4. Wording of Recommendation and Structured Consensus Procedure

Based on the systematic literature search and in coordination with the DGMKG TMJ surgery working group, a draft guideline was compiled by a steering group (C.S., A.N., T.E.), including recommendations and statements, and was distributed to the participants by e-mail. A structured consensus was achieved by means of a Delphi process [19], which, using e-mail correspondence, offers the possibility of an anonymized and blinded vote.

During this process, and taking into account the respective level of evidence, participants had the option of assigning “shall”, “should” or “may” for “Strength of Recommendation” (equivalent to Grade of Recommendation A, B and 0, respectively) or to abstain. Regarding “Statements”, the options were agreement, disagreement and abstention. Furthermore, participants had the option to submit questions, comments and advocate text alterations (as a rule, to be supported by literature reference). As were the results of the vote, these were presented anonymized in the subsequent rounds and voted on, if applicable. The results were assessed by an independent non-voting member of the steering group (Ch. S). In cases where no consensus could be achieved, and in cases where a strong consensus was expected, (see Table 4), the item to be voted on, sometimes after a minor alteration of the wording, was voted on again in a new, blinded and anonymized consensus round.

Strength of recommendation (Graduation) was determined based on evidence levels, to include criteria such as clinical experience, feasibility in everyday practice/in a variety of settings, benefit–risk analysis for those affected, suitability for the patient target group and the German health system, together with ethical, legal and economic aspects (see Figure 2).

Recommendations that could not be sufficiently supported by literature references within the meaning of “good clinical practice” (Grade of Recommendation A with evidence level IIIb, IV or V) were rated as “Expert Consensus”. The Discussion section of this publication states the relevant Level of Evidence (I–V), Grade of Recommendation (A: strong recommendation, B: recommendation, 0: open recommendation, EC: expert consensus) and Strength of Consensus (↑↑: strong consensus, ↑ consensus) for each recommendation. For Statements, evidence level and strength of consensus are stated. In view of frequent abstentions in the interdisciplinary consensus phase (e.g., regarding specific surgical aspects), abstentions were not included in the total figure based on which the consensus was calculated (e.g., agreement: 11/13, disagreement: 0/13, abstentions: 2 → 11/11 strong consensus with 2 abstentions), with a required proportion of abstentions below 50%.

The consensus process had two stages. During the first phase (Consensus Phase OMFS (K1)), a draft guideline was developed by the members of the TMJS working group and agreed on through a Delphi process. Each member of the panel had one vote of their own in this phase. The results of this initial consensus phase OMFS (K1) were further developed and modified in the second phase (interdisciplinary consensus phase (K2)) by mandated representatives of the participating specialist medical societies (especially by the rheumatology, radiology and orthodontics working groups) and finally consented interdisciplinarily by all the representatives of the participating specialist societies. In the interdisciplinary vote K2, each participating specialist society had one vote. After a structured consensus was arrived at by the Delphi process, approbation of the guideline by the board of directors of the participating scientific societies followed, and its publication online by the AWMF on 28 July 2021 (https://www.awmf.org/leitlinien/detail/ll/007-061.html, accessed on 15 February 2022).

### 2.5. Statistics

A comparison was performed between the two consensus phases (initial consensus phase OMFS (K1) vs. interdisciplinary consensus phase (K2)) and specific recommendations and statements for each disease (RA and JIA respectively) regarding the variables “consensus” (in per cent), “strength of consensus” (no consensus, simple consensus und strong consensus), “proportion of abstentions” (in per cent) and “number of rounds”. As “strength of recommendation” remained unchanged between draft stage (K1) and interdisciplinary consensus (K2), this variable was not included. For the purpose of calculations for the variables “consensus”, “strength of consensus” and “proportion of abstentions”, the Mann–Whitney test was applied. Calculations for the variable “number of rounds” were carried out by means of the chi-squared test (Fisher’s exact test). The level of significance was defined as *p* < 0.05. The statistical calculations were performed using IBM^®^ SPSS ^®^ Statistics for Windows, version 27.0. Armonk (Westchester Country, NY, USA), IBM Corporation.

## 3. Results

### 3.1. Systematic Literature Search

The search for national and international guidelines initially returned four records on the AWMF pages, which, however, proved not to be relevant to the topic. A search for individual publications returned a total of 3771 records. After selection by title, abstract and full text, incorporation of the results of a hand search and an update of sources, a total of 394 sources were included in the guideline (for details, see Figure 1: Overview systematic literature search). The initial literature search was performed in February 2018, the sources were last updated in April 2021. Another search for recommendations regarding potentially relevant new sources was performed during and for the purpose of preparation of this paper in November 2021 (PRISMA items 6 and 16a). Due to the limited evidence base (only a few systematic studies in the literature are of grade Ia and IIIa; primary sources, especially evidence level IIIb, IV and V), apart from case-control studies sources with evidence level IV (e.g., case series) and V (case reports) were included in the guideline (PRISMA item 5).

### 3.2. Consensus Process

#### 3.2.1. OMFS Consensus Phase (K1) (Initial Draft Version Consensus)

Out of the 7 members of the guideline group “TMJS” of the German Association for Oral and Maxillofacial Surgery (Deutsche Gesellschaft für Mund-, Kiefer- und Gesichtschirurgie—DGMKG) selected and requested to participate, 6 members agreed to participate in the specific working groups of the initial consensus phase (see Appendix A). From October 2018 to November 2019 and in two Delphi rounds, 18 recommendations, 6 expert consensuses and 38 statements were voted on (totalling 62 recommendations/statements).

#### 3.2.2. Interdisciplinary Consensus Phase K2

Thirteen scientific societies with 6 subgroups and 7 patient associations were selected and requested to participate in the interdisciplinary consensus phase, out of which 7 scientific societies with 5 subgroups and one patient association agreed to participate (see Appendix A). From February 2020 to March 2021, and in two Delphi rounds, 25 recommendations, 7 expert consensuses and 45 statements were voted on (totalling 77 recommendations/statements). The structured consensus process and approbation of the guideline by the board of directors of the participating scientific societies were completed in June 2021.

#### 3.2.3. Statistical Analysis of the Consensus Process

In the initial OMFS consensus phase (K1), 54/62 (87.1%) of recommendations/statements were able to achieve a strong consensus, 2/62 (3.2%) a simple consensus, while 6/62 (9.7%) were unable to achieve a consensus at all. In the interdisciplinary consensus phase, 70/77 (90.9%) of recommendations/statements resulted in a strong consensus, 3/77 (3.9%) in a simple consensus and 4/77 (5.2%) in no consensus. There were no statistically significant differences in the consensus percentages or the strength of the consensus between the initial consensus phase OMFS (K1) and the interdisciplinary consensus phase K2 (see Table 5 and Table 6). The proportion of abstentions of 12.9% in the interdisciplinary consensus phase K2, however, was significantly higher than in the initial consensus phase OMFS (K1) with 10.3% abstentions (*p* < 0.001, Table 5).

In the initial consensus phase OMFS (K1), 7/62 (11.5%) of recommendations/ statements required a second voting round. In the interdisciplinary consensus phase (K2) this proportion was 6/77 (7.6%). No statistically significant difference resulted, however (*p* = 0.559, Table 7).

There were no statistically significant differences regarding RA- and JIA-specific recommendations or statements for the variables consensus percentage, strength of consensus, proportion of abstentions and number of rounds (see Table 8, Table 9 and Table 10).

Comparison between Consensus OMFS (K1) vs. Interdisciplinary Rounds (K2):

**Table 5 jcm-11-01761-t005:** Proportion of abstentions and consensus (significant results are highlighted by bold characters).

		Proportion of Abstentions	Consensus
Consensus OMFS (K1)	Mean	0.1027	0.8562
	Standard deviation	0.23345	0.33050
Interdisciplinary Consensus (K2)	Mean	0.1288	0.9282
	Standard deviation	0.16198	0.21870
Mann–Whitney test—exact signature (2-tailed)		**<0.001**	0.227

**Table 6 jcm-11-01761-t006:** Strength of consensus.

	No Consensus	Consensus	Strong Consensus
Consensus OMFS (K1)	9.7%	3.2%	87.1%
Interdisciplinary Consensus (K2)	5.2%	3.9%	89.2%
Total	7.2%	3.5%	89.1%
Mann–Whitney test—exact signature (2-tailed)		0.489	

**Table 7 jcm-11-01761-t007:** Number of rounds.

	Round 1	Round 2
Consensus OMFS (K1)	88.5%	11.5%
Interdisciplinary Consensus (K2)	92.4%	7.6%
Chi-squared test—Fisher’s exact test—exact signature (2-tailed)		0.559

Comparison between specific statements and recommendations, JIA and RA:

**Table 8 jcm-11-01761-t008:** Proportion of abstentions and consensus.

		Proportion of Abstentions	Consensus
JIA	Mean	0.0903	0.9358
	Standard deviation	0.14482	0.15857
RA	Mean	0.1010	0.9343
	Standard deviation	0.14722	0.14943
Mann–Whitney test—exact signature (2-tailed)		0.684	0.735

**Table 9 jcm-11-01761-t009:** Strength of consensus.

	No Consensus	Consensus	Strong Consensus
JIA	4.2%	4.2%	91.7%
RA	0%	27.3%	72.7%
Total	2.9%	11.4%	85.7%
Mann–Whitney test—exact signature (2-tailed)			0.297

**Table 10 jcm-11-01761-t010:** Number of Rounds.

	Round 1	Round 2
JIA	90.2%	9.8%
RA	86.4%	13.6%
Chi-squared test—Fisher’s exact test—exact signature (2-tailed)		0.687

#### 3.2.4. Identification of Controversial Areas in the Consensus Process

The content of certain statements, recommendations and topics was subject to debate and involved a prolonged consensus process during the Delphi procedure. Consensus was defined as problematic according to the below criteria:Criterion 1: Simple consensus not achieved (agreement ≤ 75%) in at least one roundCriterion 2: Modification of text required to achieve a higher level of consensus -from approval by majority (≤75%) to simple consensus (76–95%) or from-simple consensus to strong consensus (>95%)Criterion 3: Proportion of abstentions >25%Criterion 4: Persistent dissenting vote despite modification

The Delphi process was able to identify 14 controversial recommendations or statements (14/77, equivalent to 18.2% of all recommendations and statements), of which 8 were unable to achieve a consensus (≤75%), 9 required text modifications to achieve a higher level of consensus, in 9 the proportion of abstentions was >25% and for one recommendation a dissenting vote persisted despite modification made. The problematic areas of the recommendations and statements included the below topics (for details, see Table 11):Diagnostics: Orthopantomography (OPG) for initial imaging, cone-beam computed tomography (CBCT) compared to CT, sonography for detection of active arthritis, examination of synovial fluids and histopathological surveying using Krenn scores and bone scintigraphy, biopsy of components of the masticatory muscles, electromyography and instrumental recording of movements as supplementary diagnostic methods.Therapy: Intraarticular application of corticosteroids (IACI), the use of costochondral grafts (CCG) in JIA, the NSAR concept by Nørholt and colleagues in distraction osteogenesis and LeFort I osteotomy.

## 4. Discussion

In order to avoid severe sequelae such as impaired growth, condylar resorption and ankylosis of the TMJ, timely diagnosis of chronic rheumatic arthritis of the TMJ is vital. As clinical examination alone proved insufficient [20] it is therefore to be complemented by imaging methods [IIb, A, ↑↑]. Traditional radiological methods, such as orthopantomography (OPG), CT and cone-beam computed tomography (CBCT), however, due to their limited imaging ability of soft tissue, are not an option for detection of rheumatic arthritis of the temporomandibular joint [21]. Contrast-enhanced MRI alone is able to show processes of inflammation in soft tissue in a satisfactory manner and is therefore the method of choice for diagnosis of chronic rheumatic arthritis of the temporomandibular joint [IIb, ↑↑] [22,23]. Sonography, too, is able to offer good-quality imaging of soft tissue and was therefore discussed as an alternative to MRI technology in the consensus rounds. A systematic review, however, revealed the clearly superior quality of contrast-enhanced MRI, with a limited number of studies available [24]. In addition, other than MRI, the method has been insufficiently standardized so far [13,14,25]. For these reasons, sonography has not been rated a suitable alternative to MRI for diagnosis [IIIb, ↑↑] due to limited availability of studies and pending standardization. Should the MRI findings prove insufficient regarding the involvement of osseous structures, CT and CBCT are possible alternative options [IIb, ↑↑] [21]. The latter is rated by some authors as superior regarding dosage and cost efficiency [26,27,28], this was discussed during the Delphi process. In this context, diagnostic precision of CBCT and patient exposure to radiation will depend on the choice of device and examination protocol [29] (p. 1), [30,31] (p. 1). Furthermore, in Germany, statutory health insurance does not cover the costs, and reimbursement is identical to CT [32].

Due to the inconsistent correlation between clinical findings and imaging, and the high prevalence of involvement of the temporomandibular joint in RA and JIA, identification of an appropriate screening remains a highly topical question. The use of MRI is under controversial discussion because of high cost, limited availability, the requirement for contrast agent administration and the often necessary sedation or general anaesthesia. While clinical examinations play a crucial role for initial assessment, follow-up of progress and evaluation of interventions they have their limitations for the purpose of screening without additional imaging for validation [5]. Orthopantomography (OPG) provides a low-cost medical imaging method, comparatively low radiation exposure and broad availability in dental offices [33,34]. During the consensus process, however, a number of representatives argued against the method for routine screenings for bony involvement due to insufficient display of subtler bony pathologies [35,36], and its limitation to coarser structures [37]. OPG thus provides an option for initial imaging as a basis for the diagnosis of more advanced bony lesions only [IIb, ↑↑].

Despite being a cost-efficient, low-risk and widely available method sonography offers only limited sensitivity and specificity and is therefore not a satisfactory option [24]. The superior sensitivity of bone scintigraphy provides for early detection of bone remodelling processes—however, at the expense of specificity [38]. For diagnosis and follow-up of the progress of chronic rheumatic arthritis of the temporomandibular joint, it constitutes a third-choice diagnostic tool only; and due to the radiation exposure involved, its use should be avoided for children and adolescents [V, ↑↑]. Thus, there remains a great need for a practicable screening tool for routine clinical use [IV, ↑↑] and currently, the best option is therefore a combination of clinical examination (e.g., once a year by means of the TMJaw group’s screening protocol [12]) complemented by MRI diagnostics.

In the case of borderline MRI findings with potential clinical relevance, also the use of invasive diagnostics was discussed during the consensus process, such as the option of examining synovial fluid in adult patients. This method was described initially by Alstergren et al. [39], and subsequently evaluated [40,41] and validated [42] several times. It has, however, mainly been applied for basic research in the working group around Alstergren —while reports on clinical implementation are rare [43] and indications therefore are restrictive. An examination of synovial fluid may be considered in individual cases in adult patients whose response to conservative therapy is not satisfactory, and borderline MRI findings [IIIb, 0, ↑↑]. As an alternative, there is the option to implement a histopathological assessment using the Krenn score. This method is well established, e.g., in orthopaedics for larger joints, by means of taking a synovialis sample in the context of minimally invasive or open joint surgery [44,45]. Not only can conclusions be drawn on the efficiency of pharmacological therapy based on the Krenn score, it also enables a distinction between degenerative and rheumatic joint affection [44]. Despite its validation and application in orthopaedics [46], the score has not been established in oral and maxillofacial surgery. The background is the comparative smaller sample volume available from arthroscopic biopsies as compared to orthopaedics relating to larger joints. Due to its added diagnostic value [46] a synovial analysis of the TMJ by means of the Krenn score may in individual cases be considered independent of an intervention otherwise indicated for further differential diagnostics [IIIb, 0, ↑↑]. To avoid procedures that are not absolutely necessary, indications shall be strictly limited—especially with regard to patients below 17 years of age [V, EC, ↑↑].

The fundamental basis of the therapy of chronic rheumatic arthritis of the TMJ is a systemic pharmacological therapy by means of disease-modifying antirheumatic drugs (DMARDs). It can be supplemented by measures focusing on the TMJ, such as jaw rest, soft foods, physiotherapy, bite splint (occlusal and distraction splints) and functional orthodontic appliances [2]. If the response to conservative measures is not satisfactory, minimally invasive interventions, such as arthroscopy with lavage, arthrocentesis and intraarticular corticosteroid injections (IACI) are further options. The latter has been subject to controversial discussion both in the literature and again during the consensus process. Both in the context of RA and JIA, a considerable short- and medium-term reduction of subjective symptoms and improvement of joint function could be shown together with a low rate of complications of the intervention [47,48,49]. However, damage to the articular structures up to the destruction of the TMJ has also been reported, especially after the repeat application of IACIs [50,51,52]. A systematic review has been able to show the effect of corticosteroids on the articular cartilage to be relative to dosage applied and time, with a more favourable effect of short-time use and lower dosage and a chondrotoxic effect of long-term use and higher dosage [53]. Fouda has been able to show a similar effect on the TMJ in an animal model [54], Haddad in RA patients [52] and Lochbuehler et al. in JIA patients. [55]. Apart from indications of possible damage to articular structures, heterotopic ossifications have been reported as a result of repeat IACIs into the TMJ [55,56,57], with the number of repeat IACIs as a presumable risk factor [57]. In JIA patients in childhood and adolescence, the question needs to be considered, if IACIs may affect jaw growth. In animal experiments, there has been first evidence of a negative effect on mandibular growth in healthy mice [58,59,60] and experimental animals with antigen-induced arthritis [59,60,61]. Lochbuehler et al. have been able to prove a significant reduction of TMJ growth in humans as a result of repeat IACIs [55]. Due to its potentially chondrotoxic effect, the inherent risk of heterotopic ossification and inhibited mandibular growth, IACIs are to be applied—if at all—as one-time injections [IIIb, ↑↑] in the context of chronic rheumatic arthritis and should be restricted to cases refractory to therapy [V, B, ↑↑]. The interdisciplinary consensus group unanimously agreed with recommendations in the literature [55,61], that application as a regular or continuous therapy is to be avoided [IIIb, ↑↑]. Prospectively, there might be alternatives to corticosteroids for intraarticular injections in the TMJ. Animal studies with antigen-induced TMJ arthritis and human studies with arthritic conditions other than chronic rheumatic TMJ arthritis injections with simvastatin [62,63] and platelet concentrate [64,65] showed promising anti-inflammatory effects and reduced symptoms. Further research is needed in this field.

Should all attempts at conservative and minimally invasive therapy options fail, open surgery shall be considered, as this offers the prospect for the patient to profit with regard to symptoms, functionality or aesthetics [IV, EC, ↑↑]. In adult patients whose TMJ is beyond repair, has collapsed, been destroyed or has ankylosed, total alloplastic joint replacement represents the gold standard [66]. The results of endogenous materials (sternoclavicular and costochondral grafts) in comparison are significantly inferior in an inflammatory setting [67,68]. Therefore, in adult patients, alloplastic joint replacement should currently be chosen over an autologous alternative [69] [IIIb, B, ↑↑]. In JIA patients, on the other hand, some authors advocate the use of costochondral grafts (CCG), based on the prospect of generating growth and reported low complication rates [70,71]. This resulted in discussions among the surgeons from DGMKG, triggered by the assumed use of endogenous materials in an inflammatory setting and reports about the unpredictable growth of transplants. Svenson and Adell reported excessive graft growth in 4/7 JIA patients who had received a CCG [72]. A later study showed excessive growth in seven out of twelve JIA patients with CCG (58%), resulting in facial asymmetry in five of these patients (42%) [73]. Independent of JIA, Balaji and Balaji were able to demonstrate insufficient growth in 36% of patients with CCG, no growth in 21% and excessive growth in 29% of patients [74]. The exact causalities for this unpredictability of growth have not been established [74]. There has been evidence pointing to the length (or respectively volume) of the remaining cartilage cap as a relevant factor [75,76,77], due to the interindividual variation in the proportion of germinative cells. However, a specific recommendation is not possible in this respect [78]. Individual reports of alloplastic total joint replacements (TJR) in adolescents in exceptional and especially severe cases can be found in the literature [79,80]. Nevertheless, the risk of repeat surgery and limited durability of such prostheses (15 to 25 years) and their inability to generate growth, needs to be weighed against the risks of CCG [79]. Due to the absence of an adequate alternative joint replacement for adolescent JIA patients, and despite sometimes grave side effects, reconstruction of the TMJ by means of costochondral transplant may be performed [IV, 0, ↑↑]. After such intervention, regular follow-up clinical examinations for the detection of potential complications shall be performed until completion of mandibular growth [IV, EC, ↑↑]. In exceptionally severe cases and under specific preconditions (for details, see [69,79]) an alloplastic replacement may be made [69]. In the future, modular prostheses may be able to provide an alternative to CCG in JIA patients [81].

In JIA patients with facial deformities resistant to conservative therapy (asymmetry, mandibular retrognathia, micrognathia), orthognathic surgery should be considered after completion of the growth phase, and provided the TMJ arthritis is well controlled or inactive [IV, B, ↑↑] assessed by contrast-enhanced MRI. Available options are bilateral sagittal split osteotomy (BSSO) and mandibular forward displacement, mandibular distraction osteogenesis (DO) for mandibular forward displacement, Le Fort I osteotomy for occlusal correction and repair of an anterior open bite and genioplasty for optimization of facial aesthetics [IV, ↑↑]. In view of the risk of recurrences due to a reactivated underlying disease and resulting in repeat surgery, orthognathic surgery was discussed in the OMFS (K1) consensus phase.

Isolated recurrences have been reported [82,83]; conclusive long-term data, however, are lacking for quantification of such risk of recurrence. As a rule, the assessment of disease activity is of central importance. Based on the recommendations of the interdisciplinary TMJaw working group [15], inactive TMJ arthritis (established by means of contrast-enhanced MRI) and the cessation of deterioration of facial deformity for at least a year have been defined as basic prerequisites for orthognathic surgical interventions.

Concerning the consensus process, the proportion of abstentions in the interdisciplinary phase K2 of the Delphi process was significantly higher than in the initial DGMKG-exclusive Phase K1. This was mainly due to the heterogeneous composition of the interdisciplinary group of representatives with specialists from non-surgical fields of expertise. Regarding the strength of consensus and the required number of rounds, however, there were no significant differences between the initial draft phase OMFS (K1) and the interdisciplinary phase (K2).

The main limitation of this study is the low level of evidence of the available publications (mainly non-controlled retrospective study designs) with inhomogeneous methods, small patient collectives and a high risk of systematic bias. Unequivocal statements, therefore, were not always possible. Furthermore, 7 recommendations with Grade of Recommendation “A” (strong recommendation) had to be downgraded to “expert consensus” due to insufficient evidence. The level of evidence was determined according to the AWMF rules and standards and the 2009 Oxford Criteria. With the latest update of the rules and standards 2020 ongoing, the 2011 Oxford Criteria are currently recommended [84], which, however, have not yet been applied to this guideline, as the guideline project was initiated in 2018. Another limiting factor originates in the method of the Delphi process. While it offers the advantage of great flexibility to participants, e.g., thanks to location-independent voting, it involves voting periods of 4 to 6 weeks and great organizational effort, and a lengthy process overall. Both the initial consensus phase OMFS (K1) and the interdisciplinary consensus phase (K2), including all additional fine-tuning rounds, required one year, each.

At the same time, a great advantage of the Delphi process is the high degree of systematization and anonymization of voting results and comments it provides. This draws the focus onto factual content, protects individual contributions and allows for introduction of controversial approaches. What is more, it helps prevent potential systematic bias as an effect of social interaction and group dynamics [85]. Due to limited social interaction, the quality of the questionnaires to be voted on is crucial, and opportunity for potentially fruitful discussion may be lost due to the method of the process. To increase efficiency and to promote constructive discourse in future guideline projects, a combination of initial consolidating Delphi rounds followed by a moderated, formalized and structured consensus conference (see AWMF Rules and Standards [19]) for further clarification of critical questions could be an option—especially considering the fast improvement of the digital communication infrastructure (i.e., video conferencing) in recent months due to the current conditions under the pandemic.

## 5. Conclusions

While the Delphi process may be lengthy, it proved to be a reliable and objective method for the development of a consensus. The AWMF’s S3 Guideline on RA and JIA of the temporomandibular joint constitutes another vital step towards standardization of clinical management of chronic rheumatic arthritis of the temporomandibular joint. Regular clinical screening of the temporomandibular joint in patients with RA and JIA and contrast-enhanced MRI examinations in case of conspicuous findings constitute the diagnostic basis. In therapy, intraarticular corticosteroid injections are to be applied as restrictively as at all possible. Further research and publications of a higher level of evidence are generally required in order to be able to assume a clearer stance in those areas still under controversial discussion.

## Figures and Tables

**Figure 1 jcm-11-01761-f001:**
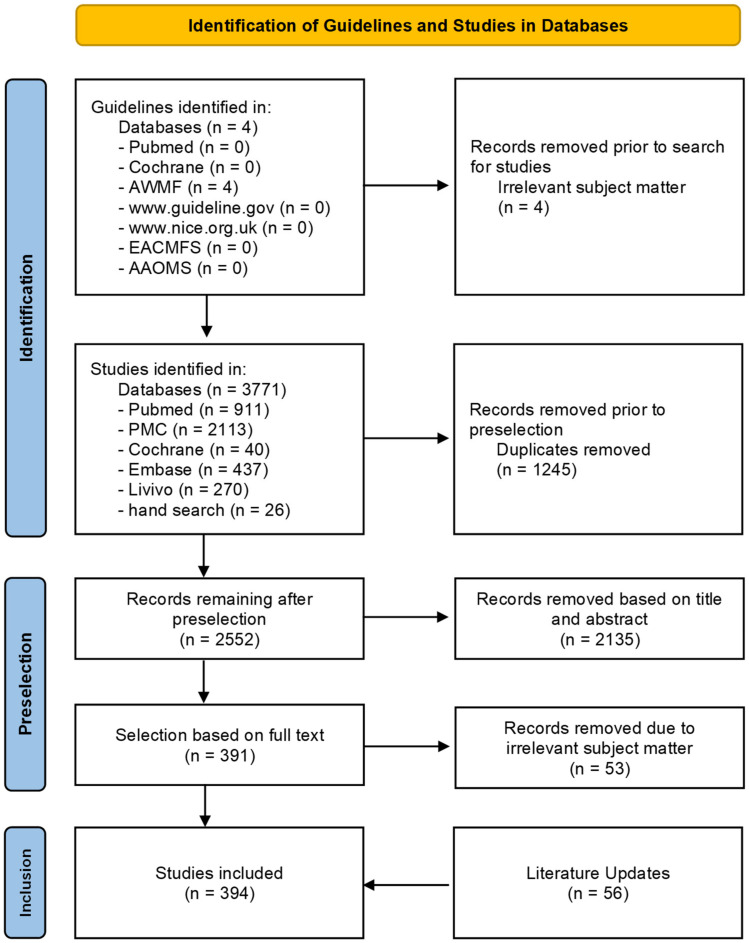
Literature search—PRISMA 2020 flow chart.

**Figure 2 jcm-11-01761-f002:**
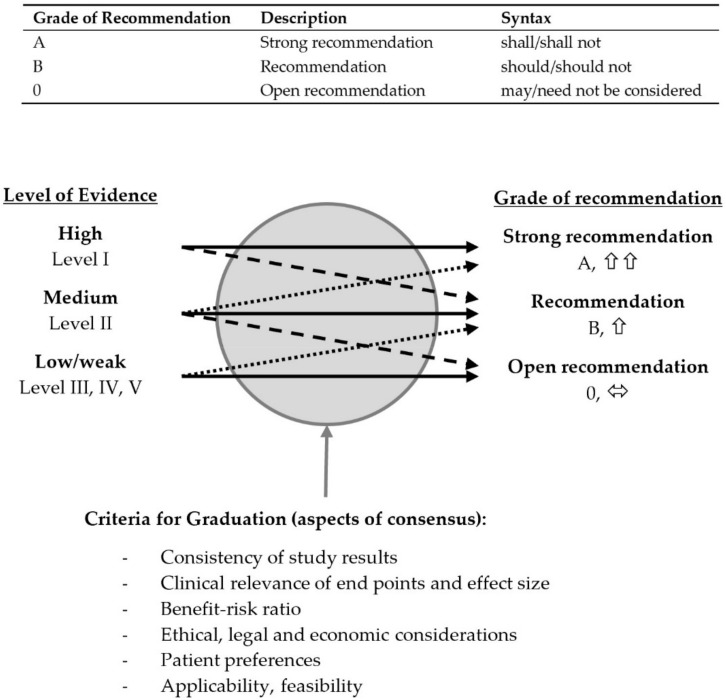
Classification of grades of recommendations according to AWMF rules and standards [19].

**Table 1 jcm-11-01761-t001:** Level of Evidence based on Oxford criteria 2009 [18].

LoE		Study Type
I	a	Meta analysis/systematic review of studies with LoE Ib
	b	“randomized controlled clinical trial” (RCT)
II	a	Meta analysis/systematic review of studies with LoE IIb
	b	“controlled clinical trial” (CCT)/comparative prospective cohort study (with control group)
III	a	Meta analysis/systematic review of studies with LoE IIIb
	b	Retrospective cohort study/case-control study
IV		Non-controlled observational study > 1 patient (e.g., case series), studies other than in vivo studies of human subjects (e.g., animal experiment, cadaver study), consensus paper
V		Case study, non-systematic secondary literature, expert opinion

**Table 2 jcm-11-01761-t002:** Assessment of methodological quality according to SIGN checklists.

Symbol	Criteria
++	High quality, overwhelming majority of criteria fulfilled (>75%), no risk or low risk of bias
+	Acceptable quality, majority of criteria fulfilled (50–75%), medium risk of bias
−	Low quality, majority of criteria not fulfilled (<50%), considerable risk of bias
0	Unacceptable, study rejected due to insufficient quality

**Table 3 jcm-11-01761-t003:** Rating of clinical relevance.

Symbol	Criteria
k++	High clinical relevance, overwhelming majority of criteria fulfilled (>75%)
k+	Acceptable clinical relevance, majority of criteria fulfilled (50–75%)
k−	Low clinical relevance, majority of criteria not fulfilled (<50%)
k0	Study without clinical relevance, study removed

**Table 4 jcm-11-01761-t004:** Classification of strength of consensus according to AWMF rules and standards [19].

Agreement	AWMF Definition
>95%	Strong consensus
76–95%	Consensus
50–75%	Approval by majority
<50%	No consensus

**Table 11 jcm-11-01761-t011:** Controversial recommendations and statements (legend: (A)/(B)/(0): Grade of Recommendation: A (strong recommendation), B (recommendation), 0 (recommendation open); LOE: Level of Evidence, K1: Initial Consensus Phase OMFS, K2: Interdisciplinary Consensus, R: Recommendation, St: Statement.

Item (Final Version)	LoE	Type	Criteria	Comment/Discussion
1. In cases of suspected chronic-rheumatic TMJ arthritis, OPG provides a cost-efficient, low-risk and widely available method for initial medical imaging for detection of advanced bony involvement of the temporomandibular joint.	IIb	St	**Criterion 2:** **K2.1 Consensus (77%), 2 comments** → *Adaptation of text, Statement added* **K2.2 Strong consensus (100%)**	Insufficient detection ability for subtler bony pathologies was pointed out, and therefore suitability as initial screening tool was questioned.
2. CBCT may be used for evaluation of bony structures of the temporomandibular joint in chronic rheumatic TMJ arthritis as potential dosage-efficient alternative to CT. Radiation exposure will largely depend on choice of device and examination protocol.	IV	R (0)	**Criterion 2:** **K2.1 Consensus (85%), 3 comments, 2/13 dissenting votes** → *Adaptation of text* **K2.2 Strong consensus (100%)**	Perceived higher dosage efficiency and cost efficiency of CBCT compared to CT questioned.
3. Sonography is currently not considered a suitable means for diagnosis and monitoring of progress of TMJ arthritis with underlying chronic rheumatic disorder due to lack of standardization and limited availability of studies.	IIIb	St	**Criterion 2:** **K2.1 Consensus (91%), abstentions 2/13 (15%), 2 Comments** → *Adaptation of text,* **K2.2 Strong consensus (100%), abstentions 2/13 (15%)**	Discussion on the role of sonography as a well-established method for the assessment of joint alterations due to arthritis in general vs. its limitations for anatomical reasons, and as not well established, for the TMJ.
4. The superior sensitivity of bone scintigraphy provides for early detection of bone remodelling processes—however, at the price of specificity. For diagnosis and monitoring of progress of chronic rheumatic arthritis of the temporomandibular joint, it is indeed a third-line choice diagnostic device. Its use should be avoided in children and adolescents due to the radiation exposure involved.	V	St	**Criteria 1,2 and 3:** **K1.1 No consensus (50%), 4/6 abstentions (67%), 1 comment** → *Adaptation of text,* **K1.2 Strong consensus (100%)**	Particular emphasis on the insufficient specificity of the method.
5. Synovialis analysis by means of the Krenn Score may, in individual cases, be considered for the purpose of further assessment and differential diagnosis, independent of an intervention otherwise indicated.	IIIb	R (0)	**Criteria 2 and 3:** **K2.1 Consensus (88%), 1 dissenting vote, abstentions 4/13 (31%), 1 comment** → *Adaptation of text, expert consensus added* **K2.2 Strong consensus (100%), abstentions 4/13 (31%)**	Discussion on the degree of invasiveness of the procedure, as it is only insufficiently established in oral and maxillofacial surgery.
6. Biopsy of components of the masticatory muscles as additional examination method is not considered a useful approach in the context mentioned above.	V	St	**Criteria 1 and 3:** **K1.1 No consensus (33%), abstentions 3/6 (50%)** **K2.2 No consensus (100%) ^2^, abstentions 9/13 (69%)**	
7. If clinical indicators point to structural damage in the absence of pain (“silent arthritis of the temporomandibular joint”) and in case of borderline MRI-diagnostic findings, extraction and examination of synovial fluid from the temporomandibular joint may be considered in individual cases in patients >17 years of age.	IIIb	R (0)	**Criteria 1, 2 and 3:** **K2.1 No consensus (66%), abstentions 3/6 (50%), 1 comment** → *Adaptation of text,* **K2.2 Strong consensus (100%)**	Discussion on the degree of invasiveness of the procedure, as it is only insufficiently established in oral and maxillofacial surgery.
8. Electromyography provides a possible additional diagnostic option.	V	St	**Criteria 1 and 3:** **K1.1 No consensus (0%), abstentions 4/6 (66%)** **K2.2 No consensus (0%), abstentions 8/13 (62%)**	
9. Instrumental recording of the movements of the mandible provides a possible additional diagnostic option.	IIb	St	**Criteria 1 and 3:** **K1.1 No consensus (0%), abstentions 5/6 (83%)** **K2.2 No consensus (29%), abstentions 6/13 (46%)**	
10. Due to the risk of severe complications (chondrotoxicity), no recommendation could be made by the guideline group in favour of IACI in cases where the temporomandibular joint exclusively is affected, or if it is intended as an additional measure during medication therapy, or for bridging during transition between medications in adults.	V	St	**Criteria 1, 2 and 3:** **K1.1 No consensus (50%), abstentions 2/6 (33%), 1 comment** → *Adaptation of text, 2 statements added* **K2.1 Consensus (88%), abstentions 5/13 (39%)** → *Adaptation of text* **K2.2 Strong consensus (100%), abstentions 5/13 (39%)**	Call for limitation to one-time application and avoidance of continuous therapy
11. Due to the risk of severe complications (disturbance of mandibular growth, heterotopic ossification), no recommendation could be made by the guideline group in favour of IACI, if it is intended as an additional measure during medication therapy, or for bridging during transition between medications in JIA.	IV	St	**Criteria 1, 2 and 3:** **K1.1 No consensus (50%), abstentions 2/6 (33%), 1 comment** → *Adaptation of text, 2 statements and 1 recommendation added* **K2.1 Consensus (90%), abstentions 3/13 (23%)** → *Adaptation of text* **K2.2 Strong consensus (100%), abstentions 3/13 (23%)**	Call for limitation to one-time application and avoidance of continuous therapy
12. In cases of loss of function of the temporomandibular joint in adolescents with JIA, otherwise refractory to therapy (e.g., ankylosis), despite sometimes grave side effects, autologous reconstruction of the TMJ may be performed by means of a costochondral graft.	IV	R (0)	**Criterion 4:** **K1.1 Consensus (83%), 1 dissenting vote, 1 comment**→ *1 statement added* **K1.2 Consensus (83%), 1 dissenting vote**	Method criticized as no longer state-of-the-art in view of side effects
13. The approach of Nørholt and colleagues, i.e., application of NSAIDs one hour prior to distractor activation and use of an occlusal splint to shift the load from the temporomandibular joint to the teeth, has been approved.	IV	St	**Criteria 1 and 3:** **K1.1 No consensus (75%), abstentions 2/6 (33%)** **K2.2 No consensus (100%) ^2^, abstentions 7/13 (54%)**	
14. For skeletal deformities in the context of TMJ involvement in JIA, or as a result of JIA, Le Fort I osteotomy for correction of occlusion and repair of an anterior open bite after the end of the growth phase, is a possible option in select cases, provided the underlying disease is inactive/well controlled or adequately managed, as otherwise there is a risk of recurrence. Furthermore, factors such as sufficient posterior airway space (PAS) and the basic dentofacial aspects of orthognathic surgery need to be considered.	IV	St	**Criterion 2:** **K1.1 Consensus (83%), 1 comment** → *Adaptation of text* **K1.2 Strong consensus (100%)**	Call for limitation to individual cases and added basic prerequisites for the intervention

^2^ (Despite a consent of > 75% no consensus was achieved due to a proportion of abstentions > 50%).

## Data Availability

No publicly archived datasets available; data can be supplied by the authors upon reasonable request.

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
