# Peer review of "Controversial Aspects of Diagnostics and Therapy of Arthritis of the Temporomandibular Joint in Rheumatoid and Juvenile Idiopathic Arthritis—An Analysis of Evidence- and Consensus-Based Recommendations Based on an Interdisciplinary Guideline Project"

_jcm, 2022, doi:10.3390/jcm11071761_

Round 1

Reviewer 1 Report

Manuscript ID: jcm-1621511

Title: Controversial Aspects of Diagnostics and Therapy of Arthritis of the Temporomandibular Joint in Rheumatoid and Juvenile Idiopathic Arthritis - An analysis of evidence- and consensus-based recommendations based on an interdisciplinary guideline project

1. What is the main question addressed by the research?

To identify controversial topics in the diagnostics and therapy of arthritis of the temporomandibular joint in rheumatoid and juvenile idiopathic arthritis that can affect guidelines development and further research.

2. Is it relevant and interesting?

The article is relevant and interesting.

3. How original is the topic?

The topic is current.

4. What does it add to the subject area compared with other published material?

The authors have collected and analyzed a great deal of recent data.

5. Is the paper well written?

Yes, the article is well written.

6. Is the text clear and easy to read?

Yes, but minor English editing is required.

7. Are the conclusions consistent with the evidence and arguments presented?

Yes, the conclusions consistent with the evidence and arguments presented but further studies are necessary to confirm authors’ hypothesis.

8. Do they address the main question posed?

Yes, the Authors addressed the main question posed.

Other comments:

English language: Minor English editing is required.

Abstract: To attract the reader's attention, please clarify the target of the article, and structure the abstract.

Introduction: This section needs some improvements. Please insert this sentence “Platelet concentrates injections in the therapy of temporomandibular joint arthritis management were found to be effective [https://doi.org/10.3390/ijms20020277]. Nevertheless, efficacy of platelet concentrate is at the center of a recent academic debate [https://doi.org/10.1016/j.joms.2018.01.012]”.

Materials and Methods: Please cite and follow PRISMA guidelines for systematic reviews

(https://www.equator-network.org/).

Results: Please follow PRISMA guidelines for systematic reviews

(https://www.equator-network.org/).

Discussion: This section was properly prepared

Conclusion: This section was properly prepared but further studies are necessary to confirm authors’ hypothesis.

Thanks for the opportunity to review this manuscript.

Author Response

Dear reviewer,

please find the answer letter in the word file attached.

Kind regards

Christopher Schmidt

Reviewer 2 Report

The manuscript here aims to identify and then make a consensus on recommandations and statements written about rheumatoid and juvenile idiopathic Arthritis of the TMJ. They made a wide review of literature to determine the primary recommandations/statements which then are evaluated by first an oral and maxillofacial jury and second a multidisciplinary one, through the Delphi process. This paper is well written and the method well described. The conclusions of the whole process are clearly exposed in the discussion.

Some abbreviations should be explained before use (e.g. PICOTS, OMFS; St and E in table 11)

Lines 478-480: " In JIA patients with facial deformities resistant to conservative therapy [...], orthognathic surgery should be considered after completion of the growth phase, and provided the TMJ arthritis is well controlled or inactive [IV, B, ↑↑]. " I completely agree with this recommandation. However a a clarification on the type of imaging method to be carried out to assess the inactive stage of the JIA would be interesting.

Table 11 seems excessive to me. It should either be simplified or be placed in supplemental material.

Line 515: p should be added to "Delhi"

Author Response

(The authors gave the same response as above.)
